# Kinesin Family Member C1 (KIFC1/HSET): A Potential Actionable Biomarker of Early Stage Breast Tumorigenesis and Progression of High-Risk Lesions

**DOI:** 10.3390/jpm11121361

**Published:** 2021-12-14

**Authors:** Nikita Wright, Zhihong Gong, Rick Kittles, Rama Natarajan, Tijana Jovanovic-Talisman, Padmashree Rida, Mark LaBarge, Victoria Seewaldt

**Affiliations:** 1Department of Population Science, City of Hope Comprehensive Cancer Center, Duarte, CA 91010, USA; niwright@coh.org (N.W.); rkittles@coh.org (R.K.); mlabarge@coh.org (M.L.); 2Department of Cancer Prevention and Control, Roswell Park Comprehensive Cancer Center, Buffalo, NY 14263, USA; zhihong.gong@roswellpark.org; 3Department of Diabetes Complications and Metabolism, City of Hope Comprehensive Cancer Center, Duarte, CA 91010, USA; rnatarajan@coh.org; 4Department of Molecular Medicine, City of Hope Comprehensive Cancer Center, Duarte, CA 91010, USA; ttalisman@coh.org; 5Novazoi Theranostics, Rancho Palos Verdes, CA 90275, USA; cgp_rida@yahoo.com

**Keywords:** breast cancer, kinesin family member C1 (KIFC1/HSET), centrosome amplification, neoplastic progression, tumorigenesis, human mammary epithelial cells (HMECs), high risk, cellular senescence, basal-like

## Abstract

The enigma of why some premalignant or pre-invasive breast lesions transform and progress while others do not remains poorly understood. Currently, no radiologic or molecular biomarkers exist in the clinic that can successfully risk-stratify high-risk lesions for malignant transformation or tumor progression as well as serve as a minimally cytotoxic actionable target for at-risk subpopulations. Breast carcinogenesis involves a series of key molecular deregulatory events that prompt normal cells to bypass tumor-suppressive senescence barriers. Kinesin family member C1 (KIFC1/HSET), which confers survival of cancer cells burdened with extra centrosomes, has been observed in premalignant and pre-invasive lesions, and its expression has been shown to correlate with increasing neoplastic progression. Additionally, KIFC1 has been associated with aggressive breast tumor molecular subtypes, such as basal-like and triple-negative breast cancers. However, the role of KIFC1 in malignant transformation and its potential as a predictive biomarker of neoplastic progression remain elusive. Herein, we review compelling evidence suggesting the involvement of KIFC1 in enabling pre-neoplastic cells to bypass senescence barriers necessary to become immortalized and malignant. We also discuss evidence inferring that KIFC1 levels may be higher in premalignant lesions with a greater inclination to transform and acquire aggressive tumor intrinsic subtypes. Collectively, this evidence provides a strong impetus for further investigation into KIFC1 as a potential risk-stratifying biomarker and minimally cytotoxic actionable target for high-risk patient subpopulations.

## 1. Introduction

Breast cancer is the leading form of cancer diagnosed among women in the United States [1]. Approximately 1 in 8 American women are expected to develop breast cancer in their lifetime [1,2]. Among women under the age of 40, Black/African-American women have higher incidence rates of breast cancer and are more likely to die from breast cancer overall than non-Hispanic White women [1]. Furthermore, Black/African-American women disproportionately present in the clinic with the most aggressive breast tumor subtypes, such as basal-like and triple-negative breast cancers (TNBC) [1,3,4].

The development of breast cancer has been shown to require a multistep process that involves a series of key transformative events [5,6,7]. These stages of progression have been clinically classified as ductal hyperplasia (DH) and atypical hyperplasia (AH), which reflect early histologically identifiable neoplastic changes; these stages are believed to progress to carcinoma in situ (CIS) and invasive carcinoma (IC), which reflect a transition to malignancy and tumor progression, respectively [8,9]. The critical molecular events underlying the development of each stage, also known as aging, have been shown to primarily involve the bypassing of three distinct tumor-suppressive barriers in human mammary epithelial cells (HMECs), including stress-associated stasis, replicative senescence, and oncogene-induced senescence [10,11,12]. Bypassing these crucial barriers, which involves loss of genomic integrity, cell cycle dysregulation, and telomerase reactivation, allows normal healthy cells to become immortalized or malignant [6,7]. However, no routinely assessable robust radiologic, pathologic, or molecular biomarkers currently exist in the clinic to predict the likelihood of a patient’s lesion progressing to a malignant state and becoming invasive or bypassing critical tumor-suppressive barriers. Additionally, no such biomarkers exist that can predict the intrinsic subtype of a transformed lesion.

Kinesin family member C1 (KIFC1/HSET), a microtubule binding protein of the kinesin-14 family, prevents the death of cells with centrosome amplification (CA), which is a hallmark of cancer [13,14]. CA (when a cell harbors three or more centrosomes or abnormally large centrosomes) has been observed in premalignant and pre-invasive lesions and postulated to drive pre-neoplastic changes in early stage lesions and tumor progression [14,15,16,17]. In HMECs and stem cells, CA was found to increase with age [18,19]. Specifically, CA has been found to increase from atypical ductal hyperplasia (ADH) to ductal carcinoma in situ (DCIS), to increase with higher DCIS grade, and to be more prevalent in TNBC versus non-TNBC tumors [20,21]. CA has also been shown to be associated with chromosomal instability (CIN) and aneuploidy in early stage breast lesions as well as being linked to cell cycle deregulation, telomere dysfunction, and cellular senescence [15,16,22,23,24,25]. Furthermore, induction of CA in non-transformed cell lines has been shown to be sufficient to induce tumorigenesis and to mimic oncogene-induced cellular invasion [26,27,28,29]. KIFC1 is upregulated upon induction of CA in cancer cells as a compensatory mechanism that assists cancer cells with extra centrosomes to avoid undergoing cell death [30]. Supernumerary centrosomes promote multipolar mitotic cellular divisions, which could result in levels of aneuploidy that could prevent the progeny cells from surviving. KIFC1 can circumvent this cell death by clustering extra centrosomes for proper cell division. Thus, KIFC1 upregulation has been linked to the survival and expansion of genomically unstable cells. Persistence of cells with loss of genomic integrity may promote the ability of pre-neoplastic cells to initiate early neoplastic changes, transformation, and progression.

KIFC1 is overexpressed and confers a poorer prognosis across various cancer types such as hepatocellular carcinoma, non-small cell lung cancer, ovarian cancer, prostate cancer, and breast cancer [31,32,33,34]. In breast cancer, high nuclear KIFC1 levels at the time of diagnosis have been associated with shorter overall and progression-free survival [35]. Breast tumors display approximately five-fold higher KIFC1 expression compared to corresponding normal tissue, and KIFC1 is specifically upregulated in estrogen receptor-negative breast tumors and TNBC [35,36]. KIFC1 expression was also found to be higher in breast cancer cell lines compared to premalignant cells, such as the MCF10A series and HMECs [35,36]. KIFC1 expression was found to be low in normal HMECs and in normal breast epithelia tissue [35,36]. However, a progressive increase in nuclear KIFC1 expression was observed from DH to ADH to DCIS and to IC and also correlated with increasing tumor grade [35].

Since nuclear accumulation of KIFC1 (a) is present in early stage lesions, (b) increases with progressive stages of neoplastic transformation, (c) is linked to key processes in malignant transformation, and (d) facilitates the evasion of cell death, we postulate that nuclear KIFC1 may be a potential biomarker of oncogenic transformation and predict the likelihood of a premalignant or pre-invasive lesion progressing in the clinic. KIFC1 can also be more easily assessed than CA among high-risk patients in the clinic through clinically facile methods such as immunohistochemistry (IHC) since CA is an organellular abnormality and not a protein like KIFC1. Furthermore, KIFC1 inhibition, which selectively targets cells harboring CA, has been shown to be an effective minimally cytotoxic anti-cancer strategy in preclinical studies [37]. This finding suggests that KIFC1 inhibitors may be a potential treatment option for high-risk patients with premalignant or pre-invasive lesions to prevent progression. Herein, we present evidence supporting the potential role of KIFC1 as a risk-stratifying biomarker for identifying high-risk patients and propose KIFC1 inhibition as an alternative, healthier treatment option for patients more likely to progress. We also encourage future investigation into this role to support the implementation of KIFC1 into routine clinical practice for high-risk patient subpopulations.

## 2. From the Beginning: Transition of Pre-Neoplastic Cells to an Oncogenic Phenotype

Why some normal or premalignant cells give rise to ADH, CIS, or IC while others do not remains an enigma. Answering this central clinical cancer question has required the study of pure populations of HMECs in vitro and in vivo that reflect each stage of neoplastic progression [7]. Many groups have analyzed HMEC populations in culture to increase our understanding of the cellular and molecular processes driving breast carcinogenesis. Normal HMECs display a finite capacity to divide in vitro. This cellular mortality is defined by a progressive cessation of cell growth in culture that results in senescence typically after 10–20 passages (60–70 population doublings) [5,38,39]. Tumor or transformed cells are able to escape or bypass senescence as a result of genetic changes (i.e., dysregulation of tumor suppressors and oncogenes) that allow these cells to override this limited growth potential, thus becoming immortalized or malignant [40]. In vitro, HMECs overcome three molecularly distinct tumor-suppressive barriers on their pathway to becoming malignant [10,11,41].

The first barrier is known as stress-associated stasis. This barrier can be regulated by the master tumor suppressors and cell cycle regulators, retinoblastoma (Rb) and p53. Normal HMECs that are able to overcome this barrier often exhibit deregulation of cell cycle control via loss of Rb and p53 function, although loss of p53 function is not required [7,42]. Loss of Rb function can result in aberrant cell proliferation and apoptosis, and loss of p53 function can result in desensitization of cells to checkpoint signals and apoptosis; thus, loss of either Rb or p53 function can facilitate bypass of senescence [7,43]. Inactivation of p16, which inhibits cyclin-dependent kinase 4 (CDK4) from phosphorylating Rb, bypasses Rb phosphorylation, leading to aberrant cell cycling [44]. P53 deregulation leads to abrogation of transcriptional activation of the G1 CDK inhibitor, p21, stimulating aberrant cellular growth [45]. Bypassing the stasis barrier has been suggested to correlate with hyperplasia/atypical hyperplasia in vivo as these aberrant cells exhibit clonal growth [12,42].

The second barrier, known as replicative senescence, is characterized by the absence of sufficient telomerase to continue replicating and is marked by critically shortened telomeres or telomere dysfunction, genomic instability, and activation of DNA damage response pathways [11,46,47]. In culture, normal HMECs undergo successive genome replication that is accompanied by gradual shortening of chromosomal ends, which can lead to cellular senescence once telomeric ends reach a critically short length [48]. However, normal or post-stasis HMECs that are able to upregulate or reactivate telomerase and stabilize telomeric end sequences can bypass this senescence barrier and become immortalized [7,49]. Bypassing telomere dysfunction has been suggested to correlate with DCIS in vivo, which often displays telomerase reactivation and genomic instability [42].

The third barrier, known as oncogene-induced senescence, is less well-characterized but known to involve telomerase activity and immortalization [41,50]. HMECs that overcome this barrier also deregulate critical oncogenes such as c-H-ras and c-myc [7,12]. Resistance to this barrier has been suggested to be critical for acquiring malignant characteristics and correlates with primary cancer in vivo [42]. When cultured HMECs that carried genes encoding SV40 large-T antigen, the telomerase catalytic subunit, and H-ras oncoprotein were xenografted into immunocompromised mice, they developed into tumors that were poorly differentiated and infiltrated adjacent tissue [51]. These findings suggest that most transformed HMECs have already bypassed this barrier and can acquire malignant features after the addition of one strong oncogene, which can confer their ability to invade in vivo or develop into IC.

## 3. Bypassing the Stress-Associated Stasis Barrier: KIFC1 and Cell Cycle Deregulation

Since the centrosome organizes microtubules for proper cell division, centrosome number fidelity is critical for equal partitioning of chromosomes into each daughter cell [52,53]. Cells burdened with extra centrosomes will often form three or more spindle poles during mitoses in lieu of a bipolar spindle [15,16]. This organellar abnormality and microtubule disorganization can result in cytoarchitectural alterations in tissue, leading to loss of cellular differentiation (anaplasia) [54]. Furthermore, this erroneous multipolar mitosis can lead to unequal segregation of chromosomes into each daughter cell and subsequently intolerable levels of aneuploidy. To avoid this error, cells will undergo mitotic arrest or mitotic catastrophe and activate apoptotic or necrotic cell death machinery [55,56]. However, KIFC1, which is a minus-end-directed microtubule binding protein that is activated by genomic instability signals, associates with the plus ends of microtubules and translocates to the nucleus, where it crosslinks and slides microtubules in an antiparallel fashion [57,58]. This antiparallel sliding allows extra centrosomes to aggregate at opposite poles of the cells to still form a “pseudo-bipolar spindle”, which results in the occurrence of lagging chromosomes [13,59,60]. This “pseudo-bipolar” spindle can thus allow premalignant or pre-invasive cancer cells to avert mitotic arrest and cell death so they can continue replicating as genomically unstable cells. This phenomenon can increase the genetic diversity among the cellular population [61]. This phenotypic heterogeneity in lesions can promote the selection of viable cellular subclones with advantageous traits and precipitate the expansion of more aggressive cellular phenotypes [61,62]. Hence, KIFC1-mediated centrosome clustering may be a key factor driving the survival of cells in benign lesions that are more likely to transform to a malignant state and progress.

CA has been shown to increase in normal HMECs with continued population doublings [63,64]. As previously discussed, normal HMECs that have bypassed the stasis barrier exhibit significant deregulation of the universal regulators of cell cycle control, Rb or p53. Centrosomes normally duplicate in the S phase of the cell cycle [65]. Activated Rb represses the E2F1-mediated gene transcription required for G1 to S phase transition, and p53 can halt cell division in the G1 phase by inhibiting CDK activity [66]. Centrosome duplication is tightly coordinated with genome duplication and cell division [16]. Thus, loss of Rb and p53 function along with downstream p16 and p21 loss has been shown to generate supernumerary centrosomes [63,67,68,69]. This induction of CA can lead to the selection of cells with upregulation of KIFC1 to prevent spindle multipolarity and death of progeny cells. KIFC1 expression was found to be higher in p53 mutant or null tumors compared to p53 wild-type tumors [70]. Inhibition of KIFC1 gene expression was shown to upregulate p21 and downregulate CDK2 to arrest cells in G2–M phases [32]. Furthermore, KIFC1 depletion in primary human fibroblast cells displayed features of senescence, such as β-galactosidase expression [71]. These findings suggest that KIFC1 may be upregulated as a result of Rb or p53 deregulation to allow genomically unstable cells to continue replicating and avoid cellular senescence. Thus, KIFC1 may play a key role in conferring normal HMECs, deregulated in Rb or p53, the ability to continue replicating and therefore successfully bypass the stasis barrier. An Rb loss-of-function signature was established in The Cancer Genome Atlas by identifying genes that correlate with E2F1 and E2F2 expression in breast cancer, and KIFC1 emerged as one of the top genes included in the signature [72].

KIFC1 may help pre-neoplastic cells bypass the stasis barrier through alternative mechanisms. KIFC1 may assist cells with CA that activate p53, as a result of causing mitotic defects, to avert apoptosis, allowing these cells to persist [61]. Additionally, KIFC1 has been shown to protect the apoptosis resistance protein survivin from degradation by E3 ligase APC/C [35]. KIFC1 overexpression in cancer cells was found to accelerate cell cycle kinetics, particularly from G2 to M phase. Specifically, KIFC1 overexpression upregulated survival signals such as phosphorylated Bcl2, Aurora B kinase, cyclin B1, D1, and A [35]. Cyclin D1 overexpression in particular has been shown to facilitate bypassing of the stasis barrier in normal HMECs [35,43].

## 4. Bypassing the Replicative Cellular Senescence Barrier: KIFC1 and Loss of Telomere Function and Genomic Stability

As previously discussed, the ability of normal or post-stasis HMECs to correct telomere dysfunction or continue replicating with critically shortened telomeres can allow this subset of cells to bypass the second and one of the most important senescence barriers to immortalization. Telomere dysfunction as a result of telomere attrition (critical shortening of chromosomal telomeres), also known as replicative senescence, has been shown to elicit genomic instability/CIN and activation of DNA damage response repair pathways. Specifically, telomere dysfunction can induce breakage–fusion–bridge cycles that can cause structural and numerical chromosomal aberrations resulting in CIN and aneuploidy [23,73].

However, evidence exists suggesting that centrosomal aberrations precede the emergence of *TP53* mutations and end-to-end fusions during early carcinogenesis [16,74]. Telomere dysfunction has been suggested to induce supernumerary centrosomes and shown to directly correlate with increased levels of CA in HMECs, cancer cell lines, and tumor tissue. Specifically, genotoxic stress-mediated telomere dysfunction via perturbation of p16 activity in post-stasis HMECs induced the presence of centriole overduplication [64]. This stimulation of telomere dysfunction via genotoxic stress promoted localization of telomerase transcriptional elements-interacting factor (TEIF), a transactivator of human telomerase reverse transcriptase subunit, to the centrosome and induced CA [24]. TEIF was found to positively correlate with CA in colorectal tumors [75]. Telomere dysfunction induced in *Drosophila* oogenesis caused deregulation of centrosome biogenesis, leading to embryonic lethality [22]. Since CA also elicits CIN and genomic instability, telomere dysfunction may cause CIN and aneuploidy to arise in pre-neoplastic HMECs or early stage lesions.

KIFC1 is possibly upregulated upon induction of CA as a result of telomere dysfunction. This upregulation can confer a survival advantage among cells with dysfunctional telomeres. Thus, KIFC1 may promote the survival of cells with loss of telomere function. Therefore, KIFC1 may act as a key player in facilitating pre-neoplastic cells that have acquired telomere abnormalities to bypass the replicative cellular senescence barrier. Inhibiting KIFC1, and subsequently declustering extra centrosomes, could potentially induce apoptosis of telomere-dysfunctional cells. KIFC1 phosphorylation induced upon DNA damaging agent treatment conferred cancer cell resistance to this therapy, and inhibition of this KIFC1 phosphorylation repressed centrosome clustering and tumor recurrence [76]. An alternative theory on KIFC1′s potential role in bypassing this senescence barrier is that through its involvement in promoting phenotypic diversity, it may be fostering the survival of subcellular clones that selectively reactivate telomerase.

## 5. Bypassing the Oncogene-Induced Senescence Barrier: KIFC1 and Ras Signaling

The mechanisms underlying oncogene-induced senescence remain poorly elucidated. However, HMECs that have attained immortality via reactivation of telomerase are not vulnerable to this senescence barrier but exhibit the acquisition of malignant properties after overexpression of Raf-1, Ras, or ErbB2 [42]. Induction of K-Ras, alone or co-expressed with c-Myc (transcription factor downstream of Ras signaling), was shown to induce CA in premalignant human mammary glands and HMECs via dysregulation of key regulators of the centrosome duplication cycle, Cyclin D1/CDK4 and Nek2 [77]. Transduction of post-stasis HMECs with c-myc was shown to promote bypassing of the replicative senescence barrier [43]. Thus, KIFC1 may be upregulated upon Ras-induced CA and thereby confer survival of these cells. Knockdown of KIFC1 was found to inhibit the MAPK signaling cascade and downstream signaling, suggesting that KIFC1 inhibition may abrogate Ras-induced CA. These findings suggest that HMECs upregulated in oncogenic Ras signaling may be bypassing the oncogenic-induced senescence barrier by upregulating KIFC1 as a mechanism to cope with Ras-stimulated CA.

The epithelial–mesenchymal transition (EMT) process confers mesenchymal characteristics that impart invasive capabilities [78]. The EMT process is marked by loss of the epithelial cell junction protein E-cadherin and upregulation of the transcriptional factors N-cadherin, Snail, and ZEB1. Downregulation of E-cadherin is necessary for KIFC1 to efficiently cluster extra centrosomes by increasing cortical contractility [79]. Thus, low clustering capacity was found to correlate with high levels of E-cadherin. Knockdown of KIFC1 increased expression of E-cadherin and decreased expression of N-cadherin, Snail, and ZEB1 in cancer cells [80]. Thus, the existence of CA and concomitant upregulation of KIFC1 in pre-invasive lesions may also be accompanied by loss of E-cadherin at cell–cell junctions, thereby additionally imparting the ability to invade surrounding tissue to these lesions. Hence, KIFC1-mediated centrosome clustering may impart invasive capabilities to pre-invasive cells via downregulation of E-cadherin.

## 6. Luminal or Basal-like: KIFC1 and Intrinsic Subtype Specification

Nuclear KIFC1 levels increase with increasing neoplastic progression and breast tumor aggressiveness. Specifically, KIFC1 expression has been demonstrated to be higher in basal-like versus luminal breast cancers and higher in TNBCs versus non-TNBCs [21,36,81,82,83]. Breast cancers in women of African descent develop at a significantly younger age, display a more aggressive disease course, and acquire more aggressive intrinsic subtypes, such as basal-like breast cancer and TNBC, compared to those in women of European descent. Thus, CA has been suspected to potentially drive this racial health disparity in breast cancer outcomes [84]. No such racial difference in centrosomal profiles has yet been established. However, recent evidence suggests that KIFC1, which is upregulated in the presence of CA, is significantly higher in Black/African-American TNBCs than White TNBCs [85]. Nuclear KIFC1 was found only in this study to be associated with poor outcomes in Black/African-American TNBC patients but not in White TNBC patients. Moreover, KIFC1 knockdown was observed to more significantly impact proliferation and migration in Black/African-American TNBC cell lines than White TNBC cell lines. Thus, KIFC1 rather than CA may be more essential to investigate among early stage or pre-invasive lesions as a predictor of aggressive malignant transformation and progression. It may also be advantageous to investigate the potential role of KIFC1 in the racial disparity in breast tumorigenesis and progression.

The route taken to bypass the stasis barrier independently influences the intrinsic subtype of transformed HMECs [43]. Specifically, bypassing stasis in normal HMECs with p16 shRNA generated aged strains with basal-like features, whereas overexpression of cyclin D1/CDK2 in normal HMECs generated aged strains with luminal-like features. Since KIFC1 expression is highly associated with basal-like features and TNBC, we postulate that high KIFC1 levels among premalignant or pre-invasive lesions may also predict the acquisition of basal-like or more aggressive subtypes in progressed lesions. Furthermore, we suspect that high levels of KIFC1 in early stage lesions in women of African ancestry may underlie higher incidence of basal-like and TNBC subtypes among progressed lesions of African compared to European descent. Further investigation into a potential racial disparity in KIFC1 levels in premalignant and pre-invasive samples may suggest a possible explanation for why individuals of African descent are more likely to develop breast tumors at a younger age and acquire aggressive tumor molecular subtypes. Given that the Black/African-American subpopulation is highly admixed, we recommend that genetic ancestry typing of premalignant and pre-invasive lesions for proportions of West African, Native American, and European ancestry would be critical to establishing KIFC1 as a potential predictive biomarker of aggressive malignant transformation and disease progression [86,87].

## 7. Future of Breast Cancer Risk Management: Evaluating and Targeting KIFC1 in High-Risk Patient Subpopulations

Current breast cancer risk management and prevention practices are improving but are still severely lacking in availability of robust risk-stratifying biomarkers and low-cytotoxic actionable targets for high-risk patients. We discuss evidence suggesting that KIFC1 could help fulfill this urgent need. However, further investigation into these claims is necessary to establish KIFC1 as a robust risk-stratifying biomarker for high-risk individuals. We review evidence for how KIFC1 may be involved in facilitating the bypassing of the three tumor-suppressive senescence-related barriers to breast carcinogenesis and progression as well as influencing subtype decisions, as depicted in Figure 1. Our discussion provides compelling evidence that KIFC1 should be considered as a biomarker of investigation into why certain lesions possess greater inclination to progress compared to other lesions.

ADH is a common benign lesion diagnosis reported among 5–20% of breast biopsies and is associated with a higher risk of becoming malignant or progressing to DCIS (4- to 5-fold) [88]. Owing to an increase in population-based breast cancer screening programs, DCIS now represents approximately 20–25% of all breast cancer diagnoses and is associated with greater risk of progressing into invasive breast cancer [89]. However, current management of these premalignant and pre-invasive lesions has been tricky with the administration of harsh, toxic, and invasive treatments such as surgical incisions, breast-conserving treatments (i.e., lumpectomies), mastectomies, radiation therapies, and endocrine therapies [88,89]. KIFC1 is not necessary for healthy cells to survive but is essential for cells with CA to persist. KIFC1 is also a critical mediator of supernumerary centrosome clustering [37]. In a genome-wide *Drosophila* screen, KIFC1 was identified as the top hit among all centrosome-clustering genes [90]. Additionally, malignant cells were found to be highly dependent on KIFC1 to persist [83]. Thus, KIFC1 inhibition offers a potentially specific and minimally cytotoxic therapeutic strategy in the clinic that is currently unavailable for patients harboring high-risk lesions. Conveniently, there are already rationally designed commercially available KIFC1 inhibitors that decluster supernumerary centrosomes such as AZ82, CW069, and PJ34 [37,91]. However, the regulatory mechanisms of centrosome clustering remain poorly understood, suggesting that additional research on the development of more effective anti-KIFC1 drugs is necessary to improve its potency and specificity in a clinical setting. It was recently discovered that blocking KIFC1 phosphorylation may serve as a more efficacious route toward inhibiting KIFC1-mediated centrosome clustering in patients [76]. However, KIFC1-mediated effects are not the only variables involved in breast tumorigenesis and progression. Thus, we also suggest that combining KIFC1 inhibitors with existing targeted therapeutic strategies or with future specific treatments may be necessary to effectively combat oncogenesis and tumor progression.

Although CA exists in premalignant and pre-invasive breast lesions, performing immunofluorescence-based scoring for CA is impractical in formalin-fixed paraffin-embedded tumor tissue, often utilized in the clinic. This is due to centrosomes often being lost outside of the plane of histological sections, and thus, they cannot be accurately counted and assigned to each individual cell [83]. Furthermore, quantitation of CA is time-consuming and burdensome in a clinical setting. Moreover, IHC is the routine standard method of evaluating biomarkers in the clinic. Hence, nuclear KIFC1 evaluation through IHC offers a more efficient and clinically feasible method to assess KIFC1 levels among high-risk patients. However, the identification of appropriate cut-offs for optimal risk stratification among early stage lesions warrants further investigation before incorporating KIFC1 into routine clinical practice as an early stage predictive biomarker. If successful, we assert that it may be useful to assess nuclear KIFC1 levels during a high-risk patient’s routine preventative check-up to identify whether the individual is likely to progress or not and would be an ideal candidate for preventative KIFC1 inhibition therapy.

Several groups have reported that the induction of supernumerary centrosomes is sufficient to elicit malignant transformation. In esophageal cancer, CA was present in the premalignant tissue of patients that progressed but nonexistent in the premalignant tissue of patients that did not progress [17]. However, some groups have recently contradicted the claim that CA alone is sufficient to induce neoplastic transformation, suggesting other factors may be necessary to facilitate CA-induced malignant transformation [92,93]. No linear correlation between CA and KIFC1 has yet been reported. However, a linear correlation was observed between centrosomal aberrations and KIFC1 dependency in TNBC cells [83]. Thus, among early stage lesions with CA, some lesions may express sufficient KIFC1 levels to survive and transform while other lesions may not, suggesting that KIFC1 may be more informative of the propensity of a lesion progressing than the magnitude of CA. Hence, KIFC1 may impart “survival of the fittest” capabilities to centrosome-amplified cells in high-risk lesions.

Future investigations of KIFC1 in representative HMEC systems that reflect neoplastic progression will be critical to provide preclinical evidence and rationale for routine clinical use of KIFC1 as a risk-predictive and pharmacologically targetable biomarker for high-risk patients. We also encourage additional investigation among patient-derived in vivo organoid models as well as among tissue specimens extracted from high-risk patients that have progressed versus not progressed. We assert that deeper investigation into this role of KIFC1 could significantly improve estimation of the malignant predisposition of pre-neoplastic lesions and enable individualized management of high-risk patients and prevention of disease progression while simultaneously illuminating the molecular mechanisms of carcinogenic progression.

## Figures and Tables

**Figure 1 jpm-11-01361-f001:**
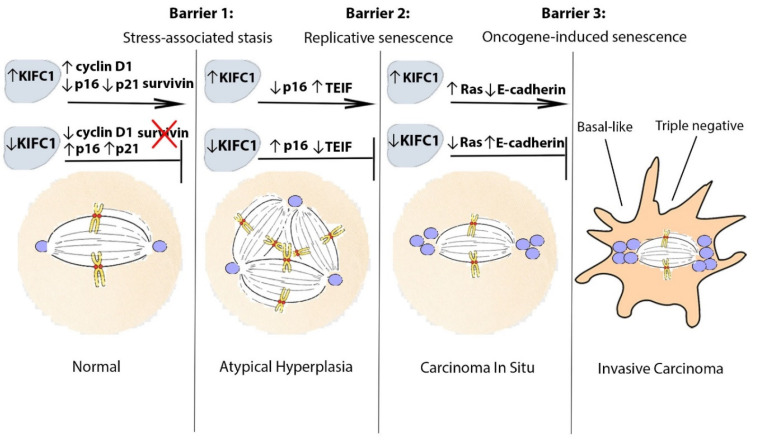
Proposed model of KIFC1-mediated breast neoplastic progression. KIFC1 may assist pre-neoplastic and pre-invasive cells to progress by facilitating their bypassing of tumor-suppressive barriers to malignant transformation and immortalization. To bypass the first barrier (stress-associated stasis), KIFC1 may upregulate cyclin D1 and/or survivin levels or be upregulated as a result of p16 and p21 loss to facilitate survival of centrosome-amplified cells. To bypass the second barrier (replicative senescence), in addition to being upregulated as a result of p16 loss, KIFC1 may be upregulated as a result of telomere dysfunction-mediated induction of TEIF to allow centrosome-amplified cells to persist. Alternatively, KIFC1 could promote selective survival of cells that can reactivate telomerase. To bypass the last barrier (oncogene-induced senescence), KIFC1 may induce the survival of cells with Ras signaling-induced CA. In addition, KIFC1 may promote the loss of E-cadherin as a result of causing cytoarchitectural reorganization, which may impart invasive capabilities to pre-invasive cells. Abbreviations: KIFC1, Kinesin family member C1; TEIF, telomerase transcriptional elements-interacting factor.

## Data Availability

Not applicable.

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
