# Peer review of "Kinesin Family Member C1 (KIFC1/HSET): A Potential Actionable Biomarker of Early Stage Breast Tumorigenesis and Progression of High-Risk Lesions"

_jpm, 2021, doi:10.3390/jpm11121361_

Round 1

Reviewer 1 Report

In this article, Wright et al. review the role of KIFC1 as a potential biomarker of high-risk lesions and early-stage breast cancer. They describe how KIFC1 might be involved in the process of bypassing 3 barriers leading to breast cancer progression along with potential future applications. This article is important as it contributes towards the global efforts for the early diagnosis of breast cancer. The article is well articulated with compelling evidence for defining KIFC1 as a potential biomarker. I would like the authors to address the following minor concerns :

  1. The authors have provided ample evidence for the role of KIFC1 in breast cancer lesions. However, most of the studies are done in HMECs. If they could provide evidence of the role of KIFC1 in other models of breast cancer including cell lines, primary cells and tissues, it would strengthen their argument. Furthermore, are there any clinical or pre-clinical studies on determining the effects of KIFC1 downregulation in vitro and in vivo and its impact on breast cancer progression?
  2. Since KIFC1 is being suggested as a biomarker, the authors also need to briefly mention how would the studies be done on patients who come for preventive check-ups.
  3. KIFC1 is demonstrated to be involved in multiple cancers. Hence, the authors need to cite those studies and mention that this biomarker is not specific to breast cancer, unless direct tissue studies are done.
  4. The figure legend should be descriptive to summarize their proposed mechanisms.
  5. On page 8, paragraph 1 the authors mention:
    1. KIFC1 is non-essential for healthy cells with no extra centrosomes but indispensable for cells with CA and an essential mediator of supernumerary centrosome clustering”. The bolded section sounds controversial and hence the authors need to tone it down and rephrase it for clarity.
    2. “Thus, KIFC1 inhibition offers a potentially specific and minimally cytotoxic therapeutic strategy in the clinic that is currently unavailable for patients harboring high-risk lesions.” Although this statement is true based on their review, it needs to be seen if it holds true in the human disease context. Furthermore, KIFC1-mediated effects are not the only issues associated with breast cancer progression. Hence, it would be better to rephrase the statement

 to suggest that it would be a better addition to combine with the existing therapeutic strategies.

  1. The authors need to define the abbreviations when they appear first: HSET, TNBC, and DCIS
  2. Proofreading of the article is essential for avoiding minor grammatical errors including spell checks, spacing, and consistency.

Author Response

Response to Reviewers:

Reviewer 1:

In this article, Wright et al. review the role of KIFC1 as a potential biomarker of high-risk lesions and early-stage breast cancer. They describe how KIFC1 might be involved in the process of bypassing 3 barriers leading to breast cancer progression along with potential future applications. This article is important as it contributes towards the global efforts for the early diagnosis of breast cancer. The article is well articulated with compelling evidence for defining KIFC1 as a potential biomarker. I would like the authors to address the following minor concerns:

Response: We are grateful for your positive and insightful comments. Please find our point-by-point responses to the minor concerns raised here:

  1. The authors have provided ample evidence for the role of KIFC1 in breast cancer lesions. However, most of the studies are done in HMECs. If they could provide evidence of the role of KIFC1 in other models of breast cancer including cell lines, primary cells and tissues, it would strengthen their argument. Furthermore, are there any clinical or pre-clinical studies on determining the effects of KIFC1 downregulation in vitro and in vivo and its impact on breast cancer progression?

Response: Great point. Additional evidence on the role of KIFC1 in other breast cancer and pre-malignant models would be help strengthen our argument. However, all evidence that exists on the role of KIFC1 in these models is described in the fourth paragraph of the introduction, where we include evidence on KIFC1 expression in normal tissue vs breast tumor tissue and normal cells vs breast cancer cell lines. We made edits to clarify this in the paragraph. We also describe how KIFC1 expression progressively increases from normal to abnormal benign to malignant tissue. More in depth preclinical studies on the role of KIFC1 in breast cancer progression is needed and we suggest this be investigated in the discussion/last section of manuscript. Actually, I am currently working in the laboratory on downregulating KIFC1 in premalignant and preinvasive in vitro and in vivo models and determining the impact on malignant transformation and tumor progression. We plan to publish this evidence soon.

  1. Since KIFC1 is being suggested as a biomarker, the authors also need to briefly mention how would the studies be done on patients who come for preventive check-ups.

Response: Great idea. We added on to the third paragraph of the last section of the manuscript to explain how this process could work with incorporating KIFC1 into routine clinical practice as a preventative biomarker.

  1. KIFC1 is demonstrated to be involved in multiple cancers. Hence, the authors need to cite those studies and mention that this biomarker is not specific to breast cancer, unless direct tissue studies are done.

Response: Yes we agree. We added updated the beginning of the fourth paragraph of the introduction section evidence supporting KIFC1’s role in multiple cancer types.

  1. The figure legend should be descriptive to summarize their proposed mechanisms.

Response: Good point. The legend should be more descriptive. We updated the figure legend to reflect this change. We also reuploaded the figure to change “oncogenic” to “oncogene” -induced senescence.

  1. On page 8, paragraph 1 the authors mention:

  1. KIFC1 is non-essential for healthy cells with no extra centrosomes but indispensable for cells with CA and an essential mediator of supernumerary centrosome clustering”. The bolded section sounds controversial and hence the authors need to tone it down and rephrase it for clarity.

Response: We agree the bold section is too strong of a claim. We toned down this statement.

  1. “Thus, KIFC1 inhibition offers a potentially specific and minimally cytotoxic therapeutic strategy in the clinic that is currently unavailable for patients harboring high-risk lesions.” Although this statement is true based on their review, it needs to be seen if it holds true in the human disease context. Furthermore, KIFC1-mediated effects are not the only issues associated with breast cancer progression. Hence, it would be better to rephrase the statement to suggest that it would be a better addition to combine with the existing therapeutic strategies.

Response: Great point. There are other variables involved in neoplastic progression. We added on to the end of the second paragraph of the discussion section that discusses this and that KIFC1 inhibitors should be combined with other strategies to more effectively prevent tumorigenesis and progression.

  1. The authors need to define the abbreviations when they appear first: HSET, TNBC, and DCIS 
  2. Response: We absolutely agree. HSET does not stand for anything and TNBC is defined in paragraph 1 of introduction. However, we defined ADH and DCIS in paragraph 3 of the manuscript.
  3. Proofreading of the article is essential for avoiding minor grammatical errors including spell checks, spacing, and consistency. 
  4. Response: Important point. We proofread this latest version for any minor grammatical errors in spelling, spacing, and consistency.

Reviewer 2 Report

This review article by Nikita et al aims to evaluate the potential of Kinesin family member 1 (KIFC1/HSET) as a biomarker for breast tumorigenesis and progression in high-risk patient population. This paper summarized the current evidence on how KIFC1 is implicated in promoting pre-neoplastic cells to circumvent three tumor-suppressive barriers to emerge to malignant and immortalized cells.

The article is well-organized, and flow of the content is logical and easy to follow. I have a few suggestions that may improve the manuscript as follows:

  1. Please define/delineate the early-stage breast tumorigenesis. Although the manuscript talked about it in the Introduction, but it is not explicit.
  2. Race is associated with biology, whereas ethnicity is associated with culture. It would be appropriate to use “racial disparity” not “ethnic disparity” and need to keep it consistent.
  3. Provide the definition for the term/abbreviation “DCIS” used in the paper.
  4. At some places, the paper makes readers confused and is hard to understand. May need to recheck syntax and split long sentence into short ones. Here are some examples:
    1. In the Introduction section: “Thus, KIFC1 upregulation has been linked to the survival and expansion of genomically unstable cells, which may promote the ability of cells with CA to initiate early neoplastic changes, transformation, and progression.”
    2. “KIFC1 is overexpressed across various cancer types and high nuclear KIFC1 levels, at time of diagnosis, have been associated with a poor prognosis in breast cancer.”
    3. “Since CA also elicits CIN and genomic instability, induction of CA upon stimulation of telomeric dysfunction could be a potential mechanism for how telomere dysfunction may cause CIN and aneuploidy to arise in pre-neoplastic HMECs or early stage lesions.”
  5. In the Introduction section, it is stated that “KIFC1 can also be more easily assessed than CA among high-risk patients in the clinic through clinically-facile methods such as immunohistochemistry (IHC).” But here it did not explain why. Then the explanation was provided in the last section. This is not logical way. This conclusive sentence should combine with the explanation in the last section.
  6. There is a typo in the paper, which make the content misleading. In the second section “From the beginning”, the authors mentioned that “Bypassing telomere dysfunction has been suggested to correlate with DCIS in vivo, which often displays short telomeres, telomerase reactivation, and genomic instability.” Here, “short telomeres” was not consistent with previous description.
  7. In the first panel of Figure 1, I assume that the authors want to express here is “survival” as “survivin” is a protein.
  8. There is an issue of use of punctuation in the subtitle “Luminal or basal-like?: KIFC1 and intrinsic subtype specification”.
  9. Based on reference 41, it should be “after overexpression of to Raf-1, Ras, or ErbB241” not “exposure” in the description “However, HMECs that have attained immortality via reactivation of telomerase are not vulnerable to this senescence barrier but exhibit the acquisition of malignant properties after exposure to Raf-1, Ras, or ErbB241.”
  10. Is there any limitations or challenges regarding to use KIFC1 as a risk stratifying biomarker?

Author Response

Reviewer 2:

This review article by Nikita et al aims to evaluate the potential of Kinesin family member 1 (KIFC1/HSET) as a biomarker for breast tumorigenesis and progression in high-risk patient population. This paper summarized the current evidence on how KIFC1 is implicated in promoting pre-neoplastic cells to circumvent three tumor-suppressive barriers to emerge to malignant and immortalized cells.

The article is well-organized, and flow of the content is logical and easy to follow. I have a few suggestions that may improve the manuscript as follows:

Response: We are grateful for your positive comments and evaluation. Please find our point-by-point responses to the minor concerns raised here:

  • Please define/delineate the early-stage breast tumorigenesis. Although the manuscript talked about it in the Introduction, but it is not explicit.

Response: Great point. Early-stage breast tumorigenesis should be defined in this manuscript to provide clarity for the reader. We included a section after the introduction, "From the beginning.." that explains the latest evidence on early steps in breast carcinogenesis.

  • Race is associated with biology, whereas ethnicity is associated with culture. It would be appropriate to use "racial disparity" not "ethnic disparity" and need to keep it consistent.

Response: Great correction. We changed anywhere in the article that says ethnic and changed it to race.

  • Provide the definition for the term/abbreviation "DCIS" used in the paper.

Response: Great observation. We defined this term in the introduction of the manuscript.

  • At some places, the paper makes readers confused and is hard to understand. May need to recheck syntax and split long sentence into short ones. Here are some examples:

Response: Great point and this is necessary to get our point across. We corrected each of the following examples in the manuscript.

  • In the Introduction section: "Thus, KIFC1 upregulation has been linked to the survival and expansion of genomically unstable cells, which may promote the ability of cells with CA to initiate early neoplastic changes, transformation, and progression."

  • "KIFC1 is overexpressed across various cancer types and high nuclear KIFC1 levels, at time of diagnosis, have been associated with a poor prognosis in breast cancer."

  • "Since CA also elicits CIN and genomic instability, induction of CA upon stimulation of telomeric dysfunction could be a potential mechanism for how telomere dysfunction may cause CIN and aneuploidy to arise in pre-neoplastic HMECs or early stage lesions."

  • In the Introduction section, it is stated that "KIFC1 can also be more easily assessed than CA among high-risk patients in the clinic through clinically-facile methods such as immunohistochemistry (IHC)." But here it did not explain why. Then the explanation was provided in the last section. This is not logical way. This conclusive sentence should combine with the explanation in the last section.

Response: Good point. We further explained why KIFC1 is more easily assessed than CA in that sentence of the manuscript.

  • There is a typo in the paper, which make the content misleading. In the second section "From the beginning", the authors mentioned that "Bypassing telomere dysfunction has been suggested to correlate with DCIS in vivo, which often displays short telomeres, telomerase reactivation, and genomic instability." Here, "short telomeres" was not consistent with previous description.

Response: Great observation. We removed "short telomeres" to avoid confusion.

  • In the first panel of Figure 1, I assume that the authors want to express here is "survival" as "survivin" is a protein.

Response: Good observation. We mentioned in the "bypassing stress-associated stasis" section that KIFC1 protect survivin from ubiquitin degradation allowing cells to avoid undergoing apoptosis. So we depicted in Figure 1 high KIFC1 levels upregulated survivin allowing pre-neoplastic cells to persist and low KIFC1 levels leads to lower levels of survivin allowing cells to undergo cell death.

  • There is an issue of use of punctuation in the subtitle "Luminal or basal-like?: KIFC1 and intrinsic subtype specification".

Response: Great point. We removed the question mark in this subtitle.

  • Based on reference 41, it should be "after overexpression of Raf-1, Ras, or ErbB241" not "exposure" in the description "However, HMECs that have attained immortality via reactivation of telomerase are not vulnerable to this senescence barrier but exhibit the acquisition of malignant properties after exposure to Raf-1, Ras, or ErbB241."

Response: Great observation. We made the suggested correction.

  • Is there any limitations or challenges regarding to use KIFC1 as a risk stratifying biomarker?

Response: More investigation is needed to fully understand the limitations and challenges of using KIFC1 as a risk-stratifying biomarker. We did mention, however, in the discussion that further investigation into developing more specific and effective KIFC1 inhibitors will be necessary. We also explained in the same paragraph that inhibiting KIFC1 alone may not be enough to prevent neoplastic progression as other variables are likely involved in carcinogenesis.